# Detection of Peanut Traces in Food by an Official Food Safety Laboratory

**DOI:** 10.3390/foods11050643

**Published:** 2022-02-23

**Authors:** Clara Tramuta, Lucia Decastelli, Elisa Barcucci, Francesco Ingravalle, Sandra Fragassi, Samantha Lupi, Daniela Manila Bianchi

**Affiliations:** 1Istituto Zooprofilattico Sperimentale del Piemonte Liguria e Valle d’Aosta—Centro di Referenza Nazionale per la Rilevazione Negli Alimenti di Sostanze e Prodotti che Provocano Allergie e Intolleranze—CReNaRiA, Via Bologna, 148, 10154 Turin, Italy; lucia.decastelli@izsto.it (L.D.); elisa.barcucci@izsto.it (E.B.); sandra.fragrassi@izsto.it (S.F.); samantha.lupi@izsto.it (S.L.); manila.bianchi@izsto.it (D.M.B.); 2Istituto Zooprofilattico Sperimentale del Piemonte Liguria e Valle d’Aosta—S.S. Biostatistica, Epidemiologia e Analisi del Rischio (BEAR), Via Bologna, 148, 10154 Turin, Italy; francesco.ingravalle@izsto.it

**Keywords:** food allergen, peanut, real-time PCR, validation

## Abstract

Food safety laboratories rely on validated methods that detect hidden allergens in food to ensure the safety and health of allergic consumers. Here we present test results for the validation and accreditation of a real-time PCR assay for the detection of peanut traces in food products. The method was tested on five classes of food matrices: bakery and pastry products, meats, ready-to-eat and dairy products, and grains and milling products. Blank samples were spiked starting with the peanut samples (*Arachis hypogaea)* at a concentration of 1000 ppm. Serial dilutions were then prepared with the DNA extracted from the blank samples to a final concentration of 0.5 ppm. The limit of detection in grains and milling products, ready-to-eat, meats, bakery and pastry products was 0.5 ppm (range, Ct 27–34) and 2.5 ppm in dairy products (range, Ct 25–34). In order to determine the exclusivity parameter of the method, the ragù matrix was contaminated with *Prunus dulcis* (almonds), *Glycine max* (soy), *Sinapis alba* (mustard), *Apium graveolens* (celery), *Allium cepa* (onion), *Pisum sativum* (peas), *Daucus carota* (carrots), and *Theobroma cacao* (cocoa); no cross-reactions were observed. The method was rated satisfactory for sensitivity (98%), specificity (100%), robustness, and repeatability and it was fully validated and accredited.

## 1. Introduction

The prevalence of food allergy in the Western world is estimated at 2% of adults and up to 8% of children [1]. In Europe, the protection of allergic consumers is guaranteed by European Union regulation (EU 1169/2011) [2], which governs a consumer’s right to information and identifies 14 food allergens that must be listed on the packaging label. In the United States, the Food Allergen Labelling and Consumer Protection Act (FALCPA) requires that the eight major food allergens, which cause approximately 90% of food hypersensitivity reactions, are contained in any packaged food be declared under a name that is readily recognizable to consumers. The food allergens include milk, eggs, fish (bass, flounder, cod), shellfish (crab, lobster, shrimp), tree nuts (almonds, pecans, walnuts), peanuts, wheat, and soy. 

The two governing rules of the polymerase chain reaction (PCR) method as it relates to labeling requirements are the general rule for the labeling of prepackaged foods and the general rule for nutrition labeling of prepackaged foods. The general rule for the labeling of prepackaged foods identifies peanuts, soybeans, wheat, shellfish, fish, eggs, milk, and tree nuts as the most problematic allergens, which are the same foods the US Food and Drug Administration (FDA) considers “major allergens”. Peanuts are used widely by the food industry in the production of peanut butter, confections, roasted peanuts, snack products, and desserts [3].

Hypersensitivity to peanuts is a growing public health problem, and the ingestion of even small amounts can trigger a severe or sometimes fatal allergic reaction. A reaction will typically occur during the first years of life; peanut allergy is usually lifelong. Symptoms may appear within seconds of ingestion, peak by 30 min or develop up to 2 h later. The target organs of a peanut-related allergic reaction are the skin, the gastrointestinal (GI), and the respiratory tract. Skin-related symptoms include urticaria, angioedema, and occasional worsening of existing eczema. GI symptoms include abdominal pain, vomiting, and/or diarrhea. Respiratory symptoms can manifest with repetitive coughing, stridor, and wheezing. Furthermore, the cardiac and the central nervous system may also be involved in the setting of anaphylactic shock, in which diminished tissue perfusion leads to cardiac arrest and syncope [4]. In an American registry of fatal food-induced anaphylaxis, 37 of the 63 fatalities recorded over a 12-year period were caused by peanut ingestion. In the United Kingdom, 10 out of 37 food allergy fatalities recorded between 1992 and 1998 were related to peanut ingestion. A two-year prospective study in a pediatric population in the UK reported three deaths, none caused by peanut, and 55 severe or near-fatal food-allergic reactions, 10 of which caused by peanut [5].

The worldwide prevalence of peanut allergy varies between 1–2% (United States, United Kingdom, Canada, Australia), 0.3–0.7% (France), and 0.2–0.6% (Denmark), while it is rare in Asia, where peanut is often absent from the list of common allergenic foods. The economic cost of food allergies, comprising peanut allergies, varies based on the study but in all cases is significant. Data from a 2012 survey estimated the economic cost of any food allergy in US children at $24.8 billion annually, or $4184 per child, with $4.3 billion in direct medical costs. Hospitalization accounted for $1.9 billion, followed by outpatient visits to an allergist ($819 million), emergency department visits ($764 million), and pediatrician visits ($543 million) [6].

To date, 16 allergenic proteins in peanut have been recognized: three seed storage proteins (SSPs, Ara h 1, Ara h 2, Ara h 3) are the major peanut allergens and are considered the markers of IgE-mediated peanut allergy. To protect allergic consumers against the risk of allergic food reactions, food safety laboratories apply validated methods to detect hidden allergens in food products. DNA-based detection methods, such as real-time PCR, together with immunologically based methods, such as enzyme-linked immunosorbent assay (ELISA), are currently used to detect hidden allergens [7,8]. Although PCR does not detect proteins with allergenic potential, there is increasing interest in developing it as a supplement, or an alternative to less sensitive immunologically based methods presently considered the methods of choice. 

The US FDA Foods Program Regulatory Science Steering Committee (RSSC) provides guidelines for validating analytical methods in the nucleic acid sequence-based analysis of food [9]. Method validation refers to the process of demonstrating or confirming that a method is suitable for its intended purpose; it is achieved by conducting experiments to determine performance characteristics and quantify method performance after initial method development and optimization. Method validation criteria may include sensitivity, accuracy, trueness, reproducibility and robustness, and precision. 

Our Institute (IZSPLV, Istituto Zooprofilattico Sperimentale del Piemonte, Liguria e Valle d’Aosta, Turin, Italy) carries out controls and research in food safety and animal health. The Food Safety Laboratory performs analysis of samples for official microbiological controls and supports agencies in risk analysis and epidemiological outbreak investigations. The laboratory was recently selected as the Italian national reference center for the detection of food allergens and substances causing food intolerance (CReNaRiA, Centro di Referenza Nazionale per la Rilevazione negli Alimenti di Sostanze e Prodotti che provocano Allergie e Intolleranze), with a mandate to develop, validate, and accredit innovative methods for the detection of hidden allergenic substances in food products. Different from European legislation regulating food microbiology testing, which establishes the analytical approach, the ISO, and the methods for setting food safety criteria, there is no international legislation that stipulates required analytical testing for hidden food allergens [10]. Furthermore, commercial tests routinely used in official or private laboratories are protected by industrial secrets or patents: since users may be granted partial access to information made public by producers in the internal validation report, they need to check the performance and validate the test to obtain accreditation.

Here we describe the optimization and the validation protocol of a real-time PCR method (RT-PCR: SPECIALfinder MC Peanut, Generon) for the detection of peanut traces in food products. 

## 2. Materials and Methods

### 2.1. Food Matrix and Sample Preparation

Five food categories were chosen to test the method on food models that mimic foods for human consumption. The model foods were: bakery and pastry products (chocolate cookies), meats (ragù), ready-to-eat (olivier salad), dairy products (yogurt), grains and milling products (rice and barley flour). 

In order to ensure the absence of peanut in the true negative matrices, the blank samples were prepared in the laboratory using raw materials coming directly from primary production or short supply chains or that had not been transformed by the food manufacturer and used as an ingredient in a different food product. When this was not possible, the ingredient was tested to exclude the undeclared presence of the target allergen. Table 1 presents the origin of the ingredients (e.g., beans and carrots for preparing the olivier salad were collected from a family vegetable garden; beef for ragù was obtained from a portion of muscle taken during slaughtering). Blank samples were prepared using disposable equipment (trays, containers, test tubes) or washed and autoclaved (tempered glass and blades, steel cutlery, beaker, and other glassware).

A portion of the true negative blank samples was spiked with power peanut (*Arachis hypogaea)* obtained by nitrogen freezing and then lyophilized. The spiked samples were contaminated to a concentration of 1000 ppm (mg/kg) to obtain high concentration spiked samples (HCSS).

### 2.2. DNA Extraction 

DNA from the blank samples (10 replicates) and the spiked samples (10 replicates) for each matrix was extracted using ION Force FAST (Generon, San Prospero, MO, Italy). Briefly, 20 mL of solution A were added to a bottle containing 5 g of sample and incubated for 1 h at 85 °C in a water bath or in a thermomixer. After centrifugation for 10 min at 10,000 rpm, the aqueous supernatant was transferred to a 50-mL bottle, and an equal amount of buffer E was added. The samples were vortexed and centrifuged for 10 min at 10,000 rpm. A set of the subnatants were added with 7.5 mL of buffer T and 20 µL of glacial acetic acid. After filtering the samples in 12-mL syringes and 0.45 micron filters (this step was omitted for the yogurt matrix), the columns on the vacuum chamber were assembled, and the vacuum pump was operated for column purification. Two successive washes were then performed using 1 mL of buffer P. The column was centrifuged for 5 min at 7000 rpm to remove ethanol residues. DNA was eluted with 100 µL of buffer D after centrifugation at 500 rpm for 30 s and at 14,000 rpm for 5 min.

### 2.3. Real-Time PCR Procedure 

DNA extracts (5 µL) were added to 15 µL of the reaction mix (RT-PCR SPECIAL finder MC Peanut, Generon, San Prospero, MO, Italy. The PCR system targets a sequence encoding rRNA with a length of 90 bp in multiple copies in cells. The amplified DNA segments were detected with a beacon probe labeled with the fluorophore FAM (peanut) and fluorophore HEX (internal amplification control) to monitor the quality of reaction performance and quencher BHQ-1.

Real-time PCR was performed on a CFX96 real-time PCR system (Bio-Rad, Richmond, CA, USA) according to the cycling protocol: initial step at 95 °C for 3 min, followed by 35 cycles of denaturation at 95 °C for 5 s, annealing at 60 °C for 10 s, and extension at 72 °C for 20 s.

### 2.4. Determination of Specificity 

Method specificity was checked in silico by comparing sequences on BLASTn of non-target allergens, animal, and vegetable species. The primers were tested for complementarity with the sequences of the following non-target substances (allergens): almond, barley, Brazil nut, buckwheat, cashew, celeriac, celery, coconut, durum wheat, einkorn, hazelnut, kamut, lupin, macadamia nut, mustard, oats, pecan nut, pine nut, pistachio, rye, sesame, soft wheat, soy, walnut; clam (*Venus gallina*); hake (*Merluccius merluccius*); lobster (*Nephrops norvegicus*); mussel (*Mytilus edulis*); prawn (*Penaeus vannamei*); salmon (*Oncorhynchus kisutch*); sea bream (*Sparus aurata*); squid (*Loligo edulis*); yellowfin tuna (*Thunnus albacares*); animal species: bison, wild boar, buffalo, cow, chicken, donkey, duck, goat, goose, horse, quail, rabbit, sheep, swine, turkey; vegetable species: apple, apricot, arugula, egg plant, banana, basil, bean, black pepper, blackberry, broccoli, brussels sprouts, black cabbage, cacao, carrot, cauliflower, chard, cherry, chestnut, chickpea, clementine, corn, cucumber, currant, fennel, fig, flax, garlic, ginger, grapefruit, grapevine, kiwi, laurel, lemon, lentil, mahaleb, marrow (zucchini), mushroom, mango, marjoram, olive, onion, orange, oregano, parsley, pea, peach, pear, pepper, pineapple, pink peppercorn, plum, pomelo, poplar, poppy, potato, radish, rapeseed, raspberry, red cabbage, rice, saffron, savoy cabbage, shallot, spinach, strawberry, sunflower, tangerine, tarragon, thyme, tomato, turnip greens. Primer specificity was then tested by amplification of the DNA extracted from the blank samples (10 replicates for each matrix, for a total of 50 samples). In order to determine exclusivity of the method, the extracted DNA of the ragù matrix contaminated with *Prunus dulcis* (almonds), *Glycine max* (soy), *Sinapis alba* (mustard), *Apium graveolens* (celery), *Allium cepa* (onion), *Pisum sativum* (peas), *Daucus carota* (carrots), and *Theobroma cacao* (cocoa) was amplified.

### 2.5. Determination of Sensitivity

A portion of the true negative blank samples was spiked with peanut powder (*Arachis hypogaea*). The spiked samples were contaminated to a concentration of 1000 ppm (mg/kg) (HCSS). Ten replicates for each matrix (total 50 replicates) of HCSS were processed for DNA extraction as described above. DNA extracted from the HCSS at the initial concentration of 1000 ppm was diluted (1000 ppm, 250 ppm, 100 ppm, 50 ppm, 25 ppm, 10 ppm, 5 ppm, 2.5 ppm, 1 ppm, 0.5 ppm) with the DNA extracted from the blank samples to a final concentration of 0.5 ppm. DNA extracts (5 µL) were added to 15 µL of the reaction mix (RT-PCR SPECIALfinder MC Peanut, Generon, San Prospero, MO, Italy). Real-time PCR was performed on a CFX96 real-time PCR system (Bio-Rad) according to the cycling protocol: initial step at 95 °C for 3 min, followed by 35 cycles of denaturation at 95 °C for 5 s, annealing at 60 °C for 10 s, and extension at 72 °C for 20 s. 

DNA was extracted from spiked ragù samples and diluted to a theoretical concentration of 0.5, 5, 50, and 500 ppm and then tested with the RT-PCR SPECIALfinder MC Peanut, Generon (San Prospero, MO, Italy) to verify the correspondence between the serial dilutions.

The Ct values obtained from the analysis were summarized by descriptive statistics (mean, standard deviation (sd), median). We also checked for differences between the Ct values from different matrices (cookies, flour, olivier salad, ragù, yogurt) using the non-parametric Kruskal—Wallis test. Single pairwise comparisons between matrices were evaluated using the two-sample Wilcoxon (Mann–Whitney) rank-sum test. A non-parametric approach instead of the ANOVA accordingly to the normality and homoskedasticity of the Ct distributions. 

Performance of the RT-PCR SPECIALfinder MC Peanut (San Prospero, MO, Italy) on unknown samples was tested on samples from an international proficiency test organized by an ISO 17043:2010-accredited institute [11]. The trial included two blind food samples to be analyzed by the participants according to internal procedures: DNA was extracted according to the procedure described above; amplification was performed following the protocol used in the validation trial on a CFX96 real-time PCR system (Bio-Rad, Richmond, CA, USA).

### 2.6. Robustness

Method robustness is usually determined by measuring its capacity to remain unaffected by small but deliberate deviations from the experimental conditions described in the protocols. Robustness was determined by changing the real-time instrument: 5 µL DNA extracts of the spiked chocolate cookies were added to 15 µL of reaction mix and amplified on a 7500 Fast Dx Real-Time PCR system (Applied Biosystems, Foster City, CA, USA) at the same temperature and time conditions used on the CFX96 real-time PCR system (Bio-Rad, Richmond, CA, USA), with the initial step at 95 °C for 3 min, followed by 35 cycles of denaturation at 95 °C for 5 s, annealing at 60 °C for 10 s, and extension at 72 °C for 20 s

### 2.7. Repeatability 

To test repeatability of the SPECIALfinder MC Peanut kit (San Prospero, MO, Italy), the relative standard deviation repeatability (RSDr, of results obtained by the same method, by the same analyst, in the same laboratory, with the same equipment, on the same samples) was statistically calculated on 10 replicates at the limit of detection (LOD) concentration for each matrix. 

## 3. Results

### 3.1. Specificity

The in silico test of the sequences of primers and probes excluded cross-reactivity with all allergens, animal and vegetable species, and with potentially co-occurring species/varieties. The results showed 100% specificity of the PCR method, as tested by amplification of the DNA extracted from the blank samples. All 50 blank samples tested negative for the target allergen; no amplification signals in FAM or amplification signals in HEX were noted. In order to determine the exclusivity parameter of the method, the ragù matrix was contaminated with *Prunus dulcis* (almonds), *Glycine max* (soy), *Sinapis Alba* (mustard)*, Apium graveolens* (celery), *Allium cepa* (onion), *Pisum sativum* (peas), *Daucus carota* (carrots), and *Theobroma cacao* (cocoa) and no cross-reactions were observed. 

### 3.2. Sensitivity

The method sensitivity was 98% (Table 2): the LOD in grains and milling products, ready-to-eat, meats, bakery, pastry products was 0.5 ppm and 2.5 ppm in dairy products. The threshold cycle (Ct) to find the lowest value that allows high fluorescence levels and the lowest possible LOD for the PCR was between 27 and 34 (matrix LOD 0.5 ppm) and between 25 and 34 (yogurt LOD 2.5 ppm) (Table 2). Real-time PCR Ct of the food matrices spiked to a concentration 10-fold the LOD is presented in Table 3. The standard curve of the cycle threshold (Ct) calculated from the serial dilutions of DNA extracted from the ragù spiked sample is shown in Figure 1.

The results of the descriptive statistics reported in Table 2 and Table 3 suggest differences both in Sd (flour is lower than the other matrices) and in mean and median (cookies often are the lowest and yogurt often is the highest).

The Kruskal–Wallis test confirms differences between matrices, both at LOD level (*p* = 0.0008) and at 10XLOD level (*p* = 0.0064).

Table 4 shows the *p*-value (*p*) obtained from the two-sample Wilcoxon (Mann–Whitney) rank-sum test for which pairwise comparisons.

Results of blind samples analyzed in the interlaboratory trial were satisfactory, according to the final report published by the organizer. The two blind samples were reported as negative and positive, respectively. The trial was designed as a qualitative approach, as required for accreditation.

### 3.3. Robustness

The robustness test showed no statistically significant discrepancies between the two instruments, while the Ct of the single reactions varied only slightly (mean Ct, 28.03; range, 28–32.07). Congruent results for all samples indicated that the method is robust. 

### 3.4. Repeatability

The assay exhibited a high degree of repeatability for the blank samples and the spiked samples contaminated at the LOD concentration. The RSDr within the test did not exceed 25%.

## 4. Discussion

Peanut ingestion is a major cause of food allergy reactions. A diet devoid of protein is the only way allergic people can prevent unwanted exposure. Correct food product labeling is fundamental for ensuring the protection of allergic consumers. Recent Italian studies on the presence of undeclared allergens in food products underscore the importance of food labeling to protect allergic individuals [12,13,14]. Regulation EU No. 1169/2011 [2] does not identify official analytical methods for the detection of undeclared food allergens, nor has the scientific community identified a universal threshold for each allergenic protein that may be considered safe for allergic consumers. The development of rapid and sensitive methods for the detection of hidden allergens in foods is therefore of significant regulatory interest. 

The two most widely used methods for the detection of food allergens are ELISA-based and PCR-based approaches [15,16,17]. Some matrices can interfere with ELISA or can cause cross-reactivity. Limitations to the use of ELISA regard cooked or heated products because the protein molecules are denatured or broken down during heat treatment, which renders the allergens undetectable by specific antibodies [18]. By comparison, PCR methods perform better in cooked products as the DNA is less sensitive to heating than proteins. Nonetheless, what matters is that it is the proteins that trigger an allergic reaction. Regardless of the approach a laboratory adopts, analytical methods for food control must be validated before their use in food control laboratories, both official and business operators. The methods must be validated before they can be applied in complex matrices such as high-fat, dairy, or multi-ingredient foods.

Here we report the test results for the validation and accreditation of a real-time PCR assay of peanut traces in complex food matrices. The technique has the advantage that it is reliable, specific, sensitive, and rapid. As such, it can be used to detect trace amounts of peanut in food products. The food matrices were carefully selected to be representative of samples commonly collected during official monitoring in consumer protection. The food samples were prepared with ingredients that might be accidentally contaminated by peanuts; this was performed to mimic the variety of food matrices and thus cover a wide range of samples. In addition, complex food matrices were selected to mimic real-world situations in a food analysis laboratory. Prior to developing the real-time PCR, we evaluated the DNA extraction yields in complex matrices with low DNA content (yogurt and olivier salad). These two matrices are known to inhibit PCR because of lipids (e.g., oil and eggs) or require ad hoc extraction protocols in order to increase their yield. We obtained satisfactory results by optimizing the extraction protocol for these matrices. The sensitivity of the real-time PCR method was a LOD of 0.5 ppm for bakery products, milling products, ready-to eat foods, and meat products. Although dairy products have a low DNA content, the real-time PCR had a LOD of 2.5 ppm in the ten replicates of this matrix class and was considered satisfactory. These limits are compatible with current international regulations, which, however, have not set a quantitative criterion for the allergens we tested in this study. 

The statistical analysis carried out to check Cts differences among various food categories confirmed the importance of a multi-matrices validation: in the case of the quantitative approach, this variability should be taken into account. 

During validation of an analytical method, evaluation of robustness is an essential step to check its capacity to remain unaffected by small deviations from experimental conditions described in the procedure. According to the US FDA Foods Program Regulatory Science Steering Committee (RSSC) [9], the type of technical instruments (brands and models) and the concentration of reagents (probe, primers, dNTPs, salts) should be tested as factors that can potentially affect test robustness. In our study and in studies on patented tests, the most important factor in robustness testing is that the thermal cycler brand or model is different from the one on which the producers optimized the test.

Official and private laboratories performing food allergen detection analysis typically use commercial tests in which specifics of method set up (e.g., primers) are protected by patents. This poses a limitation for laboratories.

## 5. Conclusions 

The real-time PCR SPECIALfinder kit for the detection of peanut traces in food products proved easy to use and provided a rapid response within less than 50 min. Method accuracy was rated satisfactory for sensitivity (98%) and specificity (100%). Our results show that the method has been sufficiently validated and accredited. Furthermore, it was found adequate for the needs of an analytical laboratory, as it meets the purpose for which it was applied.

## Figures and Tables

**Figure 1 foods-11-00643-f001:**
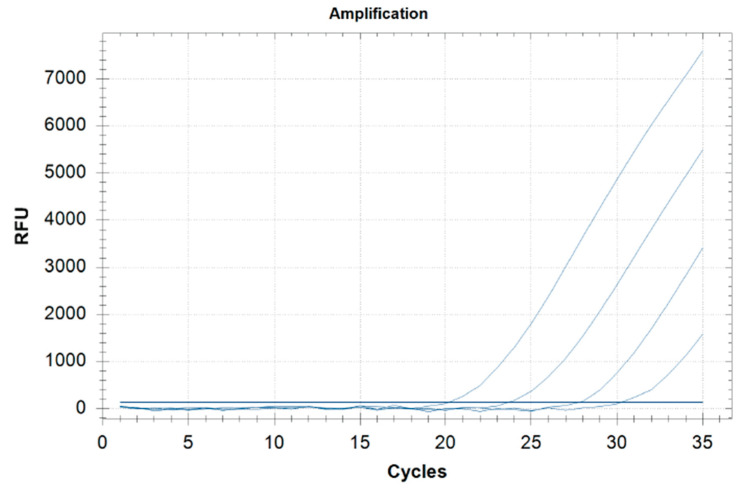
Standard curve of real-time PCR obtained by analyzing serial dilutions of DNA extracted (0.5, 5, 50, 500 ppm) from the ragù spiked samples (range, 0.5–500 ppm).

**Table 1 foods-11-00643-t001:** Origin of ingredients for preparation of blank samples.

Matrix	True Negative Food	Ingredient	Origin
Bakery and pastry products	Chocolate cookies	Corn flour	Field harvest
		Egg	Farm shell egg
		Oil	Commercial
		Sugar	Commercial
		Cocoa beans	Commercial, negative for peanuts after testing
Grains and milling products	Rice and barley flour	Rice in grains	Commercial, negative for peanuts after testing
		Barley in grains	Commercial, negative for peanuts after testing
Ready-to-eat	Olivier salad	Egg	Farm shell egg
		Oil	Commercial
		Carrot	Family vegetable garden
		Peas	Family vegetable garden
Dairy product	Yogurt	Milk	Bovine farm bulk milk tank
		Lyophilized bacterial cultures	Commercial *Lactobacillus bulgaricus* and *Streptococcus thermophilus* strains
Meats	Ragù	Beef	Muscle taken at slaughtering
		Tomato	Family vegetable garden
		Onion	Family vegetable garden
		Carrot	Family vegetable garden

**Table 2 foods-11-00643-t002:** Real-time PCR cycle threshold (Ct) for spiking food matrix to LOD.

Extraction	Cookies	Ragù	Flour	Olivier Salad	Yogurt
1	28.07	29.34	28.69	28.42	32.05
2	27.04	30.28	28.54	28.98	33.52
3	27.73	29.09	29.16	28.32	31.63
4	28.61	Not detected	30.29	30.83	30.04
5	31.27	31.24	29.37	32.36	31.27
6	28.07	39.62	29.48	27.26	25.9
7	27.71	28.38	29.27	27.58	31.24
8	28.27	29.08	28.81	28.16	31.49
9	28.04	32.04	28.54	27.62	32.38
10	28.36	28.46	28.7	29.63	32.36
Mean	28.32	29.73	29.09	28.92	31.19
Sd	1.12	1.24	0.55	1.61	2.07
Median	28.07	29.34	28.99	28.37	31.56
Minimum	27.04	28.38	28.54	27.26	25.90
Maximum	31.27	32.04	30.29	32.36	33.52

**Table 3 foods-11-00643-t003:** Real-time PCR cycle threshold (Ct) for spiking food matrix to 10XLOD.

Extraction	Cookies	Ragù	Flour	Olivier Salad	Yogurt
1	25.09	25.67	25.85	25.4	27.73
2	24.3	26.53	26.56	25.66	29.27
3	24.7	25.23	26.22	25.35	27.96
4	25.54	32.05	27.05	27.68	26.44
5	28.19	27.12	26.21	29.47	25.71
6	24.65	26.04	27.22	24.29	22.62
7	25.03	25.01	26.03	24.43	28.27
8	25.17	25.51	26	25.28	27.85
9	25.58	27.6	25.54	24.67	28.52
10	24.88	25.21	26.14	26.54	28.65
Mean	25.31	26.60	26.28	25.88	27.30
Sd	1.08	2.10	0.52	1.62	1.95
Median	25.06	25.86	26.18	25.38	27.91
Minimum	24.30	25.01	25.54	24.29	22.62
Maximum	28.19	32.05	27.22	29.47	29.27

**Table 4 foods-11-00643-t004:** Pairwise comparisons with *p*-value (*p*) and significance level of the two-sample Wilcoxon (Mann–Whitney) rank-sum test.

Pairwise Comparisons			LOD		10XLOD	
			*p*-Value (*p*)		*p*-Value (*p*)	
Cookies	vs.	Olivier salad	0.4055		0.4057	
Cookies	vs.	Flour	0.0040	**	0.0036	**
Cookies	vs.	Ragù	0.0042	**	0.0233	*
Cookies	vs.	Yogurt	0.0046	**	0.0082	**
Olivier salad	vs.	Flour	0.2263		0.0963	
Olivier salad	vs.	Ragù	0.1025		0.3643	
Olivier salad	vs.	Yogurt	0.0140	*	0.0412	*
Flour	vs.	Ragù	0.4140		0.4963	
Flour	vs.	Yogurt	0.0032	**	0.0343	*
Ragù	vs.	Yogurt	0.0199	*	0.0696	

* *p* < 0.05; ** *p* < 0.01.

## Data Availability

Not applicable.

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
