# Peer review of "Detection of Peanut Traces in Food by an Official Food Safety Laboratory"

_foods, 2022, doi:10.3390/foods11050643_

Round 1

Reviewer 1 Report

This study covers an interesting topic regarding allergen detection by using PCR. But, there are few issues needed to be clear.

1.  Provide the standard curve of cycle threshold (Ct) values calculated from serial dilutions of DNA from each sample.

2.  Methods such as sensitivity, robustness, and repeatability should be clearly described in detail.

3.  The PCR method should be fully validated in terms of linearity, precision, and accuracy for detecting allergens and verify with the sample containing unknown amounts of allergens in single and mixtures.

Author Response

thank you for your collaboration

CT

Reviewer 2 Report

This study presents the results for the validation and accreditation of a real-time PCR assay for the detection of peanut traces in food products. It is very important to investigate new protocols for the efficient detection of hidden allergens in foods.

Some observations:

  1. Line 1: Is it an “article” or communication? If it is an article, then I suggest that more references be used
  2. Line 43: In the introduction or in the Discussion section, I would suggest that some information be added:
    1. Number of deaths from peanut allergies (since peanut is the most fatal)
    2. A mention about the economic impact of hidden food allergens in foods (see product recalls in recent RASFF report)
  3. Line 118: the primers used should be mentioned
  4. Line 126: I believe, this section should be explained more at the beginning of the paragraph, exactly what was compared with what, for example that the primers used were tested for complementarity with the sequences of the non-target allergens
  5. Line 157: Why was chocolate cookies selected for this test?
  6. Line 184: In Table 2,
    1. the sensitivity 98% cannot be seen from the table, unless a standard curve correlating ppm with Ct is provided.
    2. the standard deviation should be shown
    3. a statistical analysis to show if Cts from various food categories differ would be useful
  7. Lines 205-231: this section needs some references

Author Response

thank you for your collaboration

CT

Round 2

Reviewer 1 Report

The manuscript has been improved after revising based on the comments.